# The Effect of Glyceraldehyde-Derived Advanced Glycation End Products on β-Tubulin-Inhibited Neurite Outgrowth in SH-SY5Y Human Neuroblastoma Cells

**DOI:** 10.3390/nu12102958

**Published:** 2020-09-27

**Authors:** Ryuto Nasu, Ayako Furukawa, Keita Suzuki, Masayoshi Takeuchi, Yoshiki Koriyama

**Affiliations:** 1Graduate School and Faculty of Pharmaceutical Sciences, Suzuka University of Medical Science, 3500-3 Minamitamagaki, Suzuka 513-8670, Japan; dp19001@st.suzuka-u.ac.jp (R.N.); furukawa@suzuka-u.ac.jp (A.F.); beta_amyloid@yahoo.co.jp (K.S.); 2Department of Advanced Medicine, Medical Research Institute, Kanazawa Medical University, Uchinada-machi, Ishikawa 920-0293, Japan; takeuchi@kanazawa-med.ac.jp

**Keywords:** toxic advanced glycation-end products (TAGE), glyceraldehyde (GA), Alzheimer’s disease (AD), diabetes mellitus (DM), β-tubulin, neurite outgrowth

## Abstract

Nutritional factors can affect the risk of developing neurological disorders and their rate of progression. In particular, abnormalities of carbohydrate metabolism in diabetes mellitus patients lead to an increased risk of neurological disorders such as Alzheimer’s disease (AD). In this study, we investigated the relationship between nervous system disorder and the pathogenesis of AD by exposing SH-SY5Y neuroblastoma cells to glyceraldehyde (GA). We previously reported that GA-derived toxic advanced glycation end products (toxic AGEs, TAGE) induce AD-like alterations including intracellular tau phosphorylation. However, the role of TAGE and their target molecules in the pathogenesis of AD remains unclear. In this study, we investigated the target protein for TAGE by performing two-dimensional immunoblot analysis with anti-TAGE antibody and mass spectrometry and identified β-tubulin as one of the targets. GA treatment induced TAGE-β-tubulin formation and abnormal aggregation of β-tubulin, and inhibited neurite outgrowth in SH-SY5Y cells. On the other hand, glucose-derived AGEs were also involved in developing AD. However, glucose did not make abnormal aggregation of β-tubulin and did not inhibit neurite outgrowth. Understanding the underlying mechanism of TAGE-β-tubulin formation by GA and its role in neurodegeneration may aid in the development of novel therapeutics and neuroprotection strategies.

## 1. Introduction

The balanced intake of basic nutrients including carbohydrates is necessary to maintain human health. There have been many reports regarding whether type 2 diabetes mellitus (DM) is a clinical risk factor for Alzheimer’s disease (AD), given that the incidence of AD is as much as 2–5 times higher in DM patients [1]. Reactive derivatives from non-enzymatic sugar–protein condensation reactions, as well as lipids and nucleic acids exposed to reducing sugars such as glucose, form a heterogeneous group of irreversible adducts called advanced glycation end products (AGEs) [2]. AGEs form and subsequently accumulate in various tissues during normal aging, and their formation and accumulation is highly accelerated in DM patients [3,4].

Accumulation of AGEs in the brain with aging was proposed to be involved in pathogenesis of AD [5,6]. Until recent days, glucose-derived AGE (Glu-AGE) have been mainly investigated in the AD investigation fields. However, seven immunochemically distinct classes of AGEs have been detected in sera from hemodialysis patients with diabetic neuropathy (DN-HD) [7]. Using a neuronal culture system, we previously confirmed that GA-derived toxic advanced glycation end products (toxic AGEs, TAGE) are strongly neurotoxic [8]. While the anti-TAGE antibody suppressed the neurotoxicity of serum AGEs from DN-HD patients, such effects were not observed with the other antibodies, such as glucose-AGEs [8]. In addition, we recently reported that GA intracellularly induced TAGE formation and showed neurotoxic effect within 24 h in SH-SY5Y human neuroblastoma cells and induced AD-like alterations such as tau phosphorylation, the typical phenomenon that can be seen in neurofibrillary tangles (NFTs) of AD [9]. On the other hand, TAGE contributed to various diseases, such as cardiovascular disease, nonalcoholic steatohepatitis (NASH) and cancer [7,10]. It is reported that the heat shock cognate 70 protein, heterogeneous nuclear ribonucleoprotein M and caspase-3 were identified as one of the target proteins for TAGE in NASH [11]. It is important to know TAGE-target proteins for understanding the pathological mechanism in DM-related AD. However, the earlier molecular targets of GA are not fully understood. It has been reported that AD defects in microtubule assembly and organization [12]. Microtubules regulate cell division, cell morphology and axonal stability. They comprise α/β-tubulin heterodimers that polymerize into protofilaments and associate into microtubules [13]. Understanding microtubule dynamics could provide insights into the underlying pathophysiology and present several potential therapeutic options for some neurological disorders such as AD. However, the mechanism of assembly and stabilization of microtubules by microtubules-associated proteins are still unclear. In this study, a proteomic approach was used to examine the alteration in TAGE formation in neural cells. We could identify seven candidate proteins including β-tubulin as one of the target proteins for TAGE. In AD patient brains, TAGE were mainly detected in the cytosol and axons of neurons in the hippocampus and parahippocampal gyrus, but not in senile plaques (SPs) or astrocyte [14]. Furthermore, we demonstrated that TAGE formation increased intracellular phosphorylation levels of tau, a microtubule-associated protein [9]. Thus, we focus on β-tubulin which is tau-related and microtubule-associated protein as one of the candidate-TAGE-proteins in this study. Moreover, we studied the role of TAGE-β-tubulin on neurite outgrowth in SH-SY5Y, a human neuroblastoma cell.

## 2. Materials and Methods

### 2.1. Cell Cultures

SH-SY5Y human neuroblastoma cells were purchased from ECACC (European Collection of Cell Cultures, Porton Down, UK). Dulbecco’s Modified Eagle Medium (DMEM) was obtained from Sigma-Aldrich (St. Louis, MO, USA). GA was from Nacalai Tesque, Inc. (Kyoto, Japan). All other chemicals were purchased from Wako Pure Chemical Industries, Ltd. (Osaka, Japan). SH-SY5Y cells were cultured in DMEM, containing 10% fetal bovine serum, 100 U/mL of penicillin and 100 μg/mL of streptomycin in humidified atmosphere of 95% air and 5% CO_2_ at 37 °C.

### 2.2. MTT Assay

Cell death was estimated using a 3-(4,5-dimethylthiazol-2-yl)-2,5-diphenyltetrazolium bromide (MTT) assay. An aliquot (20 μL) of 2.75 mg/mL MTT in phosphate-buffered saline was added to 200 μL of each culture medium as described previously [15]. The reaction mixtures were incubated at 37 °C for 3 h, prior to adding 200 μL HCl/isopropanol. The resultant formazan was measured by its absorbance at 550 nm using a plate reader (Model 680; Bio-Rad Laboratories, Hercules, CA, USA).

### 2.3. Preparation of Anti-TAGE Antibody

Immunoaffinity-purified anti-TAGE antibody was prepared as described previously [16]. The immunoaffinity-purified anti-TAGE antibody did not recognize well-characterized AGE structures, such as *N*-(carboxymethyl) lysine (CML), *N*-(carboxyethyl) lysine, pyrraline, pentosidine, argpyrimidine, imidazolone, glyoxal-lysine dimers methylglyoxal-lysine dimers and GA-derived pyridinium. Furthermore, the antibody did not recognize AGEs of unknown structure, such as Glu-AGEs, fructose-derived AGEs, etc. [16,17] but rather specifically recognized unique unknown TAGE structures.

### 2.4. Slot Blotting Analysis for TAGE

Cells were harvested and homogenized after being treated within 12 h with GA. An equal amount of protein was applied to a Hybri-SLOT apparatus (Gibco BRL) and transferred to a nitrocellulose membrane (Whatman, Tokyo, Japan) by vacuum filtration. After blocking with 3% bovine serum albumin for 1 h at room temperature, samples were incubated with the anti-TAGE antibody at 4 °C overnight, followed by incubation with an anti-rabbit IgG antibody (Sigma Aldrich Saint Louis, MO, USA). Antibody-bound protein bands were detected using a BCIP-NBT Kit and densitometrically analyzed.

### 2.5. Detection of TAGE Proteins by 2-Dimensional Gel Electrophoresis (2-DE) and Western Blotting

After treatment with GA, SH-SY5Y cells were collected and homogenized with lysis buffer [7M urea, 2M thiourea, 4% CHAPS, 30 mM Tris, supplemented with 1% protease inhibitor cocktail], then the homogenate was centrifuged at 30,000× *g* at 4 °C for 30 min and the supernatant collected. TAGE proteins were detected by 2-DE as described previously [18]. Immobiline Dry Strips (18 cm, pH 4–7; GE Healthcare, Tokyo, Japan) were rehydrated with rehydration buffer at room temperature overnight. The first-dimension (isoelectric focusing; IEF) was run using Cool PhoreStar IPG-IEF (Anatech, Tokyo, Japan). After 2-DE of 10% SDS-polyacrylamide gel electrophoresis was run, the proteins were transferred to a nitrocellulose membrane (GE Healthcare, Tokyo, Japan) and TAGE proteins were detected using primary rabbit anti-TAGE antibody (1:500), secondary HRP-conjugated goat anti-rabbit IgG antibody (Sigma Aldrich), and ECL western blotting detection reagents (GE Healthcare, Tokyo, Japan). 

### 2.6. Protein Identification

Protein samples from control or GA-treated cells were subjected to 2-DE as described above and the spots were manually excised from the 2D gels stained with Coomassie Brilliant Blue R-250 (BioRad, Tokyo, Japan). The gel pieces were destained with 50% acetonitrile in 25 mM ammonium bicarbonate at room temperature, then desalted with 100% acetonitrile and then 50 mM ammonium bicarbonate. The gel pieces were digested with 10 μg/mL trypsin (Promega, Madison, WI, USA) solution at 37 °C overnight. The extracted peptides were subjected to matrix-assisted laser desorption/ionization time of flight tandem mass spectrometry (MALDI-TOF/TOF MS). The data were submitted to a MASCOT search engine for identification (https://www.matrixscience.com).

### 2.7. Western Blot Analysis

Cell samples treated with vehicle or GA were extracted, then the samples and tubulin proteins (porcine brain, Cosmobio, Tokyo, Japan) were subjected to polyacrylamide gel electrophoresis using a 5%–20% gradient gel as previously described [9]. Cell extracts were boiled with buffer solution (0.02% bromophenol blue, 3% sodium dodecyl sulfate, 2-mercaptoethanol, 30% glycerol, 30 mM Tris-HCl) for 5 min. The separated proteins were transferred to a nitrocellulose membrane and incubated with primary anti-β-tubulin (1:500, D3U1W CST, Tokyo, Japan), anti-β-actin (1:500, Gene Tex, San Antonio, TX, USA), and secondary antibodies (Sigma-Aldrich). Protein bands were detected using a BCIP/NBT kit (Funakoshi, Tokyo, Japan). Protein bands isolated from cells cultured under various conditions were analyzed densitometrically using Scion Image software (Scion Corp. Frederick, MD, USA). All experiments were repeated at least three times.

### 2.8. Neurite Outgrowth

SH-SY5Y cells were differentiated with 1% fetal bovine serum to avoid overgrowth of cells and retinoic acid (RA) at 10 μM for 24 h. The effects of GA were observed following addition to the culture medium for 12 h during RA-induced differentiation. Neurite outgrowth stained with anti-α-internexin was observed by fluorescence microscopy and assessed by Scion Image software. Neurite outgrowth was quantified by obtaining 20 random images per dish from five independent dishes and assessing the longest neurites in each image (*n* = 100) [19]. We defined the control axon length to 100% and the average neurite length was expressed as the mean ± SEM. We did experiments with masking test by two coauthors (R.N. and Y.K.).

### 2.9. Immunocytochemistry

Cells were cultured and fixed in 0.1% glutaraldehyde containing phosphate saline buffer (pH 7.4). The cells were blocked with Blocking One (Nacalai Tesque) and incubated with primary anti-β-tubulin (1:500, D3U1W, CST) and anti-α-internexin (1:500, Abcam, Tokyo, Japan, ab10830) antibodies. The cells were then incubated with Alexa fluoro 488 anti-IgG (Molecular Probes, Eugene, OR, USA) and the cell nuclei were stained with acridine orange (AO, Dojindo, Tokyo, Japan). To confirm there were no immunoreactivities of non-specific or auto fluorescence, we used a secondary antibody as a negative control.

### 2.10. Statistics

All results are reported as mean ± SEM. Differences between groups were analyzed using one-way ANOVA, followed by Dunnett’s multi-comparison test with PASW Software (SPSS Inc., Chicago, IL, USA). *p* values < 0.05 were considered statistically significant.

## 3. Results

### 3.1. Detection and Identification of TAGE Proteins

In our previous study, 1 mM GA increased the formation of TAGEs in SH-SY5Y cells and showed cell death within 24 h [9], but it was unclear which proteins were the targets of TAGE formation. GA treatment induced neurotoxicity within 24 h (Figure 1A) and TAGE formation was significantly observed from 12 h (Figure 1B). We comprehensively detected TAGE protein before cell death, by 2D immunoblot analysis with anti-TAGE antibody at 12 h treatment of GA. The pattern of TAGE proteins differed between GA-treated for 12 h and vehicle control SH-SY5Y cells (Figure 1C,D). Individual spot volumes represent the amount of TAGE protein levels. In the GA-treated cells, the intensities of seven spots increased (data not shown). To identify these proteins, MALDI-TOF/TOF MS analysis were performed by searching a sequence database using specific peptide mass data (Appendix A). We identified several protein spots by amino acid sequence analysis, including human β-tubulin (MW: 50 kDa, pI: 4.78) (Figure 1C,D, arrows).

### 3.2. TAGE Formation and Abnormal Aggregation of β-Tubulin by GA

We examined the formation of TAGE by incubating recombinant tubulin proteins with 0, 0.3, 0.7, or 1 mM GA for 12 h. GA dose-dependent increased the formation of TAGE (Figure 2A–D). Interestingly, the levels of 55 kDa monomer tubulin protein (Figure 2A,D) increased, and lower polymer bands of 154 kDa (Figure 2A,C), upper band of 272 kDa (Figure 2A,B) appeared. To confirm GA-dependent β-tubulin aggregation, we performed a western blot using an anti-β-tubulin antibody (Figure 2E–H). The monomer band dose-dependently decreased after GA-treatment (Figure 2E,H) whereas all polymer β-tubulin bands become denser in a dose-dependent manner (Figure 2E,G; lower band, Figure 2E,F; upper band). Next, we confirmed the formation of TAGE-β-tubulin by GA treatment in SH-SY5Y cells (Figure 2I–M). We used β-actin antibody for confirmation of loading control in Figure 2I. As the band size is similar between β-actin (Figure 2I) and β-tubulin monomer band (Figure 2J), we used the same lot samples with another loading membrane (Figure 2I,J). Non-specific bands were seen because we used large quantities of protein (180 μg) in Figure 2I to indicate aggregation of β-tubulin bands clearly in Figure 2J. The monomer band significantly decreased after GA-treatment (Figure 2J,M) whereas all polymer β-tubulin bands (Figure 2J,L; lower band, Figure 2J,K; upper band) become denser after GA-treatment.

### 3.3. GA Inhibits Neurite Outgrowth From SH-SY5Y Cells

As β-tubulin is present in growing neurites as well as in cell bodies with various localization profiles, we observed the localization of β-tubulin using an immunocytochemical study. In the first, we checked non-specific immunofluorescence using only secondary antibody (Figure 3A,B). SH-SY5Y cells have short neurite in vehicle treatment (Figure 3A). Both auto fluorescence and non-specific fluorescence staining could not be seen in Figure 3B. In the vehicle control, β-tubulin positive staining was seen in the growth cone, which is involved in stabilizing the microtubule population and axonal growth (Figure 3C). In differentiated axons treated with RA, strong staining of β-tubulin was observed in the axons and growth cones (Figure 3D). However, in RA-differentiated cells treated with GA, β-tubulin staining was observed in the cytosol and in the axon hillock area, which might be a starting point for neurite outgrowth (Figure 3E). Furthermore, we evaluated neurite length by immunocytochemistry of α-internexin, which is one of the axon marker proteins [20]. We investigated whether GA suppresses neurite outgrowth in differentiated SH-SY5Y cells. At 10 μM, RA significantly induced neurite outgrowth (Figure 3G,I) compared with a vehicle treatment (Figure 3F,I) whereas 1 mM GA inhibited neurite outgrowth by RA in SH-SY5Y cells (Figure 3H,I). The cell death of SH-SY5Y treated with RA and/or GA was evaluated using an MTT assay. RA and/or GA treatment did not have any effect on cell viability relative to the vehicle control (Figure 3J).

### 3.4. Glucose Did Not Induce Abnormal β-Tubulin Aggregation and Did Not Inhibit Neurite Outgrowth

To know whether GA specifically induced abnormal β-tubulin aggregation and inhibited neurite outgrowth, we further checked these effects by glucose. We examined the aggregation of β-tubulin by incubating recombinant tubulin proteins with 1 mM glucose for 12 h. The levels of β-tubulin were detected by anti-β-tubulin antibody (Figure 4 A–C). The levels of the monomer bands (Figure 4B, 55 kDa) and lower bands (Figure 4C, 154 kDa) of β-tubulin were detected by western blot using an anti-β-tubulin antibody. Glucose treatment did not affect the levels of both monomer and lower bands of β-tubulin compared to vehicle treatment. Next, we evaluated neurite length by immunohistochemistry of α-internexin. We investigated whether glucose changed neurite outgrowth in differentiated SH-SY5Y cells. At 10 μM, RA significantly induced neurite outgrowth (Figure 4E,G) compared with a vehicle treatment (Figure 4D,G). Using 1 mM glucose did not change neurite outgrowth by RA in SH-SY5Y cells (Figure 4F,G).

## 4. Discussion

The present study shows four salient findings: 1) β-tubulin is one target of GA-induced AGEs; 2) GA-induces abnormal β-tubulin aggregation in a dose-dependent manner; 3) TAGE-β-tubulin formation inhibits neurite outgrowth in SH-SY5Y cells; 4) glucose did not induce abnormal β-tubulin aggregation and did not inhibit neurite outgrowth. Recent epidemiological studies have reported that the risk of developing AD is higher in DM patients (2–5 fold higher compared with the non-diabetic population). The Rotterdam study surveyed over 6000 patients and indicated a strong relationship between DM and AD, with a relative risk (RR) of 1.9 [21]. Given recent interest in the relationship between insulin and AD, it is noteworthy that patients in that study receiving exogenous insulin therapy were at the highest risk (RR 4.3) of developing dementia [22,23]. AGE levels were previously shown to be increased in the brains of diabetic patients with AD [24], and thus these reports may partly explain the clinical link between DM and AD. A relationship between AGEs and AD was suggested in several reports in 1994–1995 [5,24,25]. Furthermore, glucose-derived AGEs have effects similar to amyloid β (Aβ), namely increased neurotoxicity and glucose consumption by SH-SY5Y cells [26,27]. We previously reported [20,28] that α-hydroxyaldehyde (GA and glycolaldehyde) and dicarbonyl compounds such as glyoxal, methylglyoxal and 3-deoxyglucosone contribute to the glycation of proteins. We also reported that TAGE are stronger neurotoxins rather than Glu-AGE in a neuronal culture system [8,29]. In addition, the neurotoxic effects of serum AGEs from diabetic patients on hemodialysis were eliminated by addition of an anti-TAGE-specific antibody, but not by antibodies towards glycolaldehyde-, methylglyoxal-, glyoxal-, 3-deoxyglucosone- or Glu-AGEs [8,29]. We recently reported that the epitope structure recognized by anti-TAGE antibody is different from the previously reported GA-derived AGE structures, i.e., 3-hydroxy-5-hydroxymethyl-pyridinium compound (GLAP) and triosidines structures. We found that the anti-TAGE antibody differed from antibodies for well-defined AGEs as well as those for AGEs derived from reducing sugar/carbonyl molecules with unknown structures [16,17]. TAGE are more neurotoxic than Glu-AGEs and CML, two extensively examined AGEs species. We also reported that TAGE may be general causative agents for the development of neurodegenerative diseases such as AD [9]. On the other hand, a Glu-AGE antibody was shown to react with SPs mainly with the amyloid core, whereas the GA-AGE antibody showed no immunoreactivity with SPs [14]. These results suggest that Aβ may be glycated by glucose rather than GA. TAGE are mainly present in the neurons of the hippocampus and parahippocampal gyrus and are mainly localized in the cell body of neurons [14]. Glu-AGEs were detected in both intracellular and extracellular sites, whereas TAGE were only found intracellularly, indicating that the mechanism underlying the neurotoxicity induced by Glu-AGEs and TAGE is different. Glycation is a post-translational modification produced by a reaction between reducing carbohydrates and the amino groups, such as lysine. Recently, more interest has been paid to lysine glycation from many researchers working on metabolism. However, the systematic identification of a glycation site is still challenging because the glycated residues do not show significant patterns. In this study, GA made several β-tubulin molecules aggregate. We do not know which sites of β-tubulin are modified by GA treatment, however there are at least more than 20 lysine residues in its amino acid sequence. Further studies are needed to clarify the mechanism behind the formation of abnormal aggregation of β-tubulin. Interestingly, it was reported that β-tubulin is also glycated by glucose in a DM experimental model [30] although glucose did not trigger abnormal aggregation of β-tubulin or inhibition of neurite outgrowth under our experimental conditions (Figure 4).

GA is derived from two distinct pathways: the glycolytic pathway (glycolysis) and the fructose metabolism (fructolysis) pathway [31]. Under hyperglycemic conditions, an increase in intracellular glucose stimulates the polyol pathway to generate fructose in insulin-independent tissues, including brain and nerve tissue [32]. Fructose is phosphorylated to fructose-1-phosphate and then catabolized to dihydroxyacetone phosphate and GA by aldolase B [33], and then GA promotes the formation of TAGE. Moreover, it has been reported that aldolase B is not expressed in the rat brain [34], and fructose-1-phosphate cleavage (aldolase) activity has been detected in the human brain [35]. However, it is thought that the progression of AD will take over 10 years because the enzyme expression level is considered to be very low. Although quantification of GA is currently impossible, our previous data showing that TAGE could be detected in the brain of AD patient supports this theory [14]. 

AD is characterized pathologically by the presence of NFTs at intracellular sites. NFTs are composed of paired helical filaments (PHFs) and straight filaments. The major component of PHFs is the microtubule-associated protein, tau [36,37]. Tau in PHFs shows distinctive properties such as high aggregation, hyper-phosphorylation and other post-translational modifications [38]. Tau proteins are enriched in normal neuronal axons where they regulate microtubule stability. However, tau is detached from microtubules and aggregates in the cytosol in the presence of NFTs in a diseased brain. From these results, free-tau proteins may result in self aggregation and tend to be phosphorylated at disease-associated sites [39]. In our previous report, GA increased intracellular tau phosphorylation levels in SH-SY5Y cells. Microtubules are constructed by the polymerization of dimers of α- and β-tubulin by self-assembly. They serve as architectural elements and support the elongated shape. Thus, we focused on the microtubule-related and tau-associate β-tubulin protein. In this study, GA significantly inhibited neurite outgrowth by RA (Figure 3H,I). These results might implicate that abnormal aggregation of β-tubulin by GA cannot form normal heterodimers with α-tubulin and might inhibit polymerization of microtubules. This may be the reason that β-tubulin staining was observed in the axon hillock area by GA treatment in differentiated SH-SY5Y cells (Figure 3E). The inhibition of RA-induced neurite outgrowth by GA was not dependent on cell toxicity (Figure 3J). Further studies are needed in order to elucidate the exact mechanisms underlying β-tubulin aggregation by GA. 

## 5. Conclusions

Although the exact structure and/or mechanisms of TAGE-β-tubulin and its downstream signaling pathway currently remain unclear, we found β-tubulin is one of the targets of TAGE. We also demonstrated GA but not glucose-induced abnormal aggregation of TAGE-β-tubulin or inhibited neurite outgrowth. TAGE-β-tubulin may be a useful target for understanding the mechanism of DM-related AD.

## Figures and Tables

**Figure 1 nutrients-12-02958-f001:**
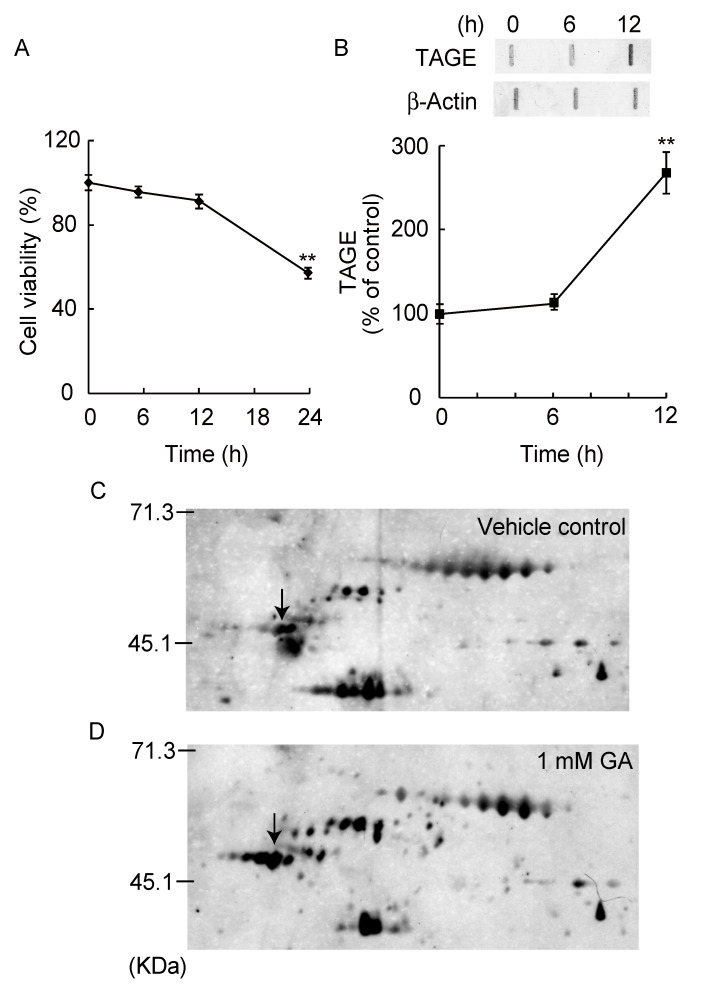
Two-dimensional difference gel electrophoresis (2D-DE) map showing positions of TAGE-protein spots. Glyceraldehyde (GA) time-dependently induced cell death in SH-SY5Y cells (**A**). Formation of toxic advanced glycation end products (TAGE) by GA treatment within 12 h. ** *p* < 0.01 vs. 0 h treatment (*n* = 6). (**B**) TAGE were measured by slot blotting analyses with an immunopurified anti-TAGE antibody. Graphical representation of TAGE bands in the slot blot. ** *p* < 0.01 vs. 0 h treatment (*n* = 3). Two-dimensional electrophoresis map showing the positions of TAGE proteins from SH-SY5Y cells treated with vehicle control (**C**) or 1 mM GA for 12 h (**D**). Differentially expressed proteins marked with black arrows are β-tubulin (MW: 50 kDa, pI: 4.78).

**Figure 2 nutrients-12-02958-f002:**
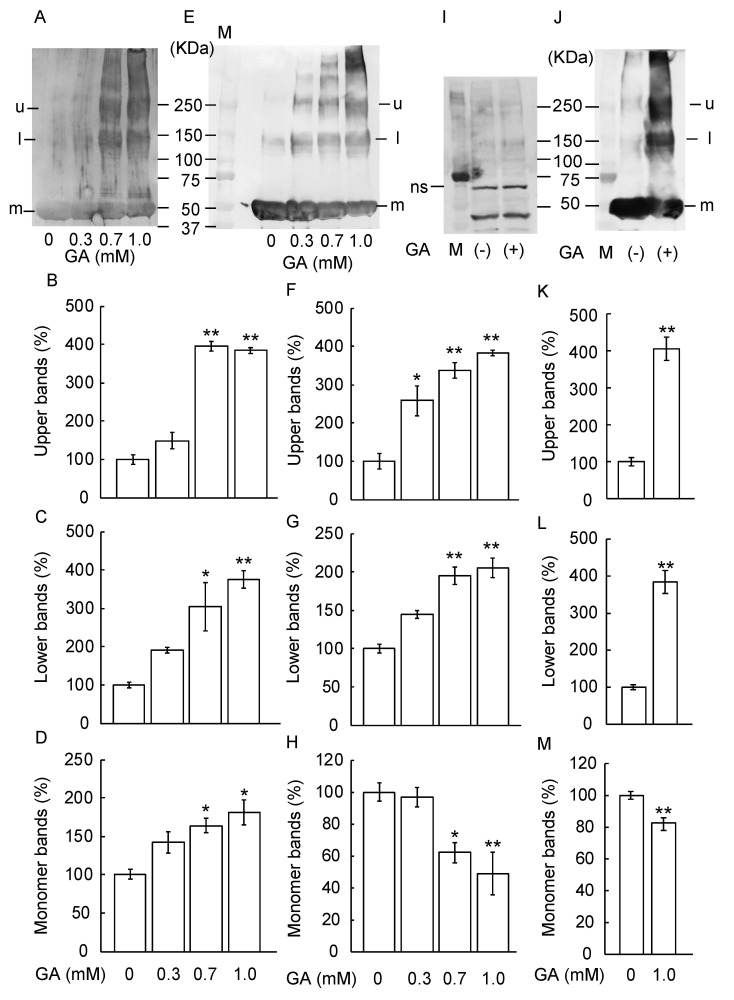
β-Tubulin dose-dependently increased abnormal aggregation by GA. (**A**–**H**) Recombinant tubulin proteins were exposed to 0, 0.3, 0.7, or 1 mM GA for 12 h. M: Marker, u: upper band, l: lower band, m: monomer band. (**A**–**D**) The levels of TAGE formation were detected by anti-TAGE antibody. (**B**) The levels of the upper bands (u) of TAGE band at 272 kDa, (**C**) The levels of the lower bands (l) of TAGE band at 154 kDa. (**D**) The levels of the monomer bands (m) of TAGE of the 55 kDa). (**E**–**H**) The levels of β-tubulin aggregation were detected by anti-β-tubulin antibody. (**E**) Western blot data by using anti-β-tubulin antibody. (**F**) The levels of the upper bands of the GA-treated β-tubulin band. (**G**) The levels of the lower bands of the GA-treated β-tubulin band. (H) The levels of the monomer bands of GA-treated β-tubulin. (**I**) SH-SY5Y cells were cultured with vehicle or 1 mM GA for 12 h. The levels of β-actin as loading control were measured by western blot analysis. ns: non-specific bands, M: marker. (**J**–**M**) SH-SY5Y cells were cultured with vehicle or 1 mM GA for 12 h. The levels of β-tubulin were measured by western blot analysis. (**J**) Abnormal β-tubulin aggregation was seen in GA-treated SH-SY5Y cells. (**K**) The levels of the upper bands of the β-tubulin in GA-treated SH-SY5Y cells. (**L**) The levels of the lower bands β-tubulin in the GA-treated SH-SY5Y cells. (**M**) The levels of the monomer bands of GA-treated SH-SY5Y cells. * *p* < 0.05, ** *p* < 0.01 versus 0 mM.

**Figure 3 nutrients-12-02958-f003:**
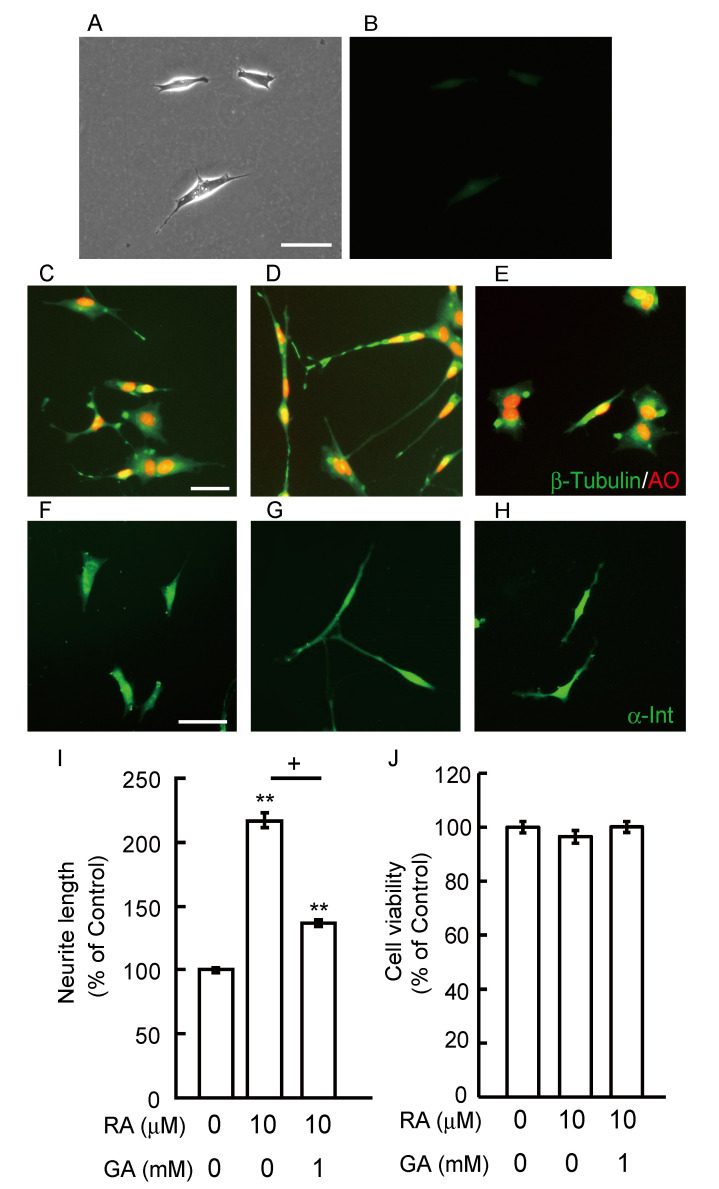
GA inhibited retinoic acid (RA)-induced neurite outgrowth in SH-SY5Y cells. (**A**) Phase contrast image in SH-SY5Y cells. Scale bar = 10 μm. (**B**) Non-specific immunoreactivity and auto immunofluorescence were checked by using a secondary antibody. (**C**–**E**) Immunocytochemical staining for β-tubulin in (**C**) vehicle-treated cells, scale bar = 10 μm. (**D**) RA-treated cells, and (**E**) RA- plus GA-treated cells. AO: acridine orange. (**F**–**H**) Photomicrographs of neurite outgrowth in SH-SY5Y cells. (**F**) Vehicle group. Scale bar = 10 μm. (**G**) RA at 10 μM, (**H**) RA plus GA. (**I**) Quantification of neurite outgrowth. ** *p* < 0.01 vs. vehicle treatment, ^+^
*p* < 0.01 vs. RA alone (*n* = 100). (**J**) Quantification of cell viability at 12 h after GA treatment (*n* = 6).

**Figure 4 nutrients-12-02958-f004:**
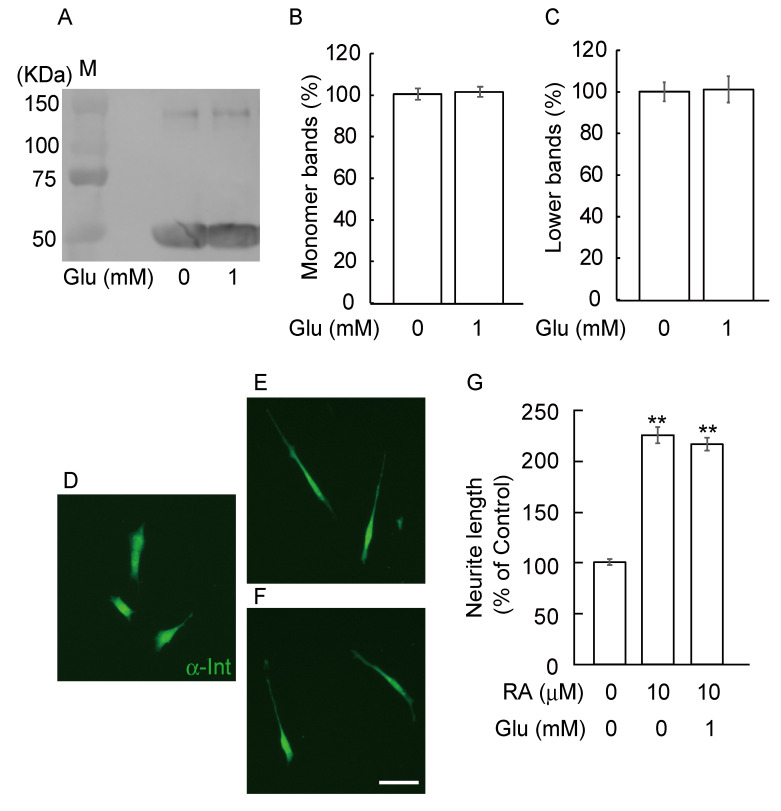
Glucose did not induce abnormal β-tubulin aggregation and not affect neurite outgrowth induced by RA. Recombinant tubulin proteins were exposed to 1 mM glucose for 12h. (**A**) The levels of β-tubulin were detected by anti-β-tubulin antibody. M: Marker. (**B**, **C**) The levels of the monomer bands (B, 55 kDa) and lower bands (C, 154 kDa) of β-tubulin. Glucose treatment did not affect the levels of both monomer and lower bands of β-tubulin compared to vehicle treatment. (**D**–**F**) Photomicrographs of neurite outgrowth in SH-SY5Y cells. (**D**) Vehicle group; (**E**) RA at 10 μM; (**F**) RA plus glucose. Scale = 10 μm. (**G**) Quantification of neurite outgrowth. ** *p* < 0.01 versus vehicle treatment (*n* = 100).

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
