# Peer review of "The Effect of Glyceraldehyde-Derived Advanced Glycation End Products on β-Tubulin-Inhibited Neurite Outgrowth in SH-SY5Y Human Neuroblastoma Cells"

_nutrients, 2020, doi:10.3390/nu12102958_

Round 1
Reviewer 1 Report
Review of: The effect of glyceraldehyde-derived advanced glycation end-products on β-tubulin-inhibited neurite outgrowth in SH-SY5Y human neuroblastoma cells by Nasu et al.
The authors assessed protein alterations in SH-SY5Y neuroblastoma cells resulting from Glyceraldehyde (GA) treatment. The authors used two-dimensional gel electrophoresis and mass spec to identify modified proteins. The authors hypothesize that: “We hypothesized that TAGE-dependent posttranslational modification of cytoskeletal proteins would induced microtubule destabilization and/or neurologic dysfunction in susceptible neurons in AD patients.” The author summarize their main findings as follows: “1) β-tubulin is one of target of GA-induced AGEs; 2) GA-induces abnormal β-tubulin aggregation in a dose-dependent manner; 3) TAGE-β-tubulin formation inhibits neurite outgrowth in SH-SY5Y cells; 4) Glucose did not induce abnormal β-tubulin aggregation and not inhibit neurite outgrowth.”
The question being asked is notable and warrants extensive research as it can provide insight into the pathological mechanisms in Alzheimer’s disease and other neurological disorders. However, it was unclear how the authors developed their hypothesis and why was b-tubulin the only target followed up on. The mass spec data is needed to validate the reasoning behind just assessing the effects of GA on tubulin. It is also not clear why the hypothesis refers to human AD patients when the model being used is by no means a specific model of AD. The data presented does not necessarily support the definitive conclusions the authors are claiming.
Major points:
- The authors need to demonstrate that 12-hr treatment of cells with GA results in the formation and/or accumulation of TAGEs and demonstrate that these are different from the prototypical AGEs by probing western blots with both antibodies in parallel—without this, it is impossible to conclude that the effects are due to TAGEs and not AGEs or some other glycated protein.
- The authors reference that 24hr exposure to TAGE is toxic to these cells. How do the authors know that the observed effects at 12hr are not just the reflection of dying cells? E.g. neurite length?
- How was loading controlled for in figure 1? There appears to be differential loading. For all Westerns, a loading control from the same membrane is required. Also, please show the replicates not just representative bands
- Mass Spec data is not provided and thus it is unclear why the authors focused on tubulin. Please provide the Mass Spec data—why was tubulin the only target selected/identified? What happens to actin, other proteins etc?
- Negative controls for immunohistochemistry is required
- Detailed description of the cell culture is necessary. Just stating that cells were differentiated with RA is not sufficient.
Minor comments:
- The authors state: “Over 100 cells were quantified and each 114 data point corresponded to the average of five independent dishes”—please provide all relevant bar graphs as independent data points to allow the reader to better understand the variability in these data.
- Please indicate how representative images were selected? Which negative controls were used? Please provide negative control images and indicate exactly how the images were quantified.
Author Response
We appreciate the comments and suggestions made by the reviewers. The reviewers’ feedback helped us to greatly improve the manuscript. We incorporated the changes suggested by the reviewers into our manuscript and figures. All revised and new text is shown blue. Our point-by-point responses to the reviewers are provided below. Because we have image in the letter, please see the attachment.
Yoshiki Koriyama

Reviewer 2 Report
In their manuscript “The effect of glyceraldehyde (GA)-derived advanced glycation end-products (AGE) on β-tubulin-inhibited neurite outgrowth in SH-SY5Y human neuroblastoma cells” the authors Nasu et al., identify β-tubulin as a toxic AGE (TAGE) protein in GA treated SH-SY5Y neuroblastoma cells, by using 2D electrophoresis, TAGE antibodies, and mass spectrometry. They further describe that GA treatment causes multimerization of β-tubulin, and inhibition of neurite outgrowth. The experiments described are straight forward, convincing, and the results are novel. Whereas I do not see the need for additional experiments, the methodological description needs to be more detailed at some points.
Major points:
Methods need some more details/clarifications.
- Cell culture medium. Was the DMEM supplemented with FCS or anything else?
- Sample preparation for WB. Which, if any reducing reagents were used?
- How long were cells treated with retinoic acid and/or GA? Were retinoic acid and GA applied together at the same time point?
- The authors write that an “average of five independent dishes” were analyzed. But that neurite length was calculated from “three independent experiments”. Later in the statistics they write “3-5 experiments”. Please clarify and provide n-numbers for independent experiments in the figure legends.
Minor points:
- How did the authors calculate the size of the monomers? I respectfully question the possibility to know their exact size down to 0.1 kDa and suggest to round the numbers, e.g. 155 and 270.
- Figure 1C, if done repetitively should be quantified for each band. Even though the β-tubulin multimerization by GA was confirmed with the data provided in Fig. 2, it would be good to show that it is true in vitro in repeated independent experiments. The authors could also try to quantify the total levels of all bands combined. Which technically might not be feasible if one of the bands is overexposed due to highly different concentrations.
- Related to Figure 1C. Results states “To confirm that the increase in the amount of TAGE on β-tubulin by GA treatment was not the increase in protein expression level, we…”. To further exclude the change in total β-tubulin mRNA levels should be analyzed. If this is not an option the sentence should be rephrased, as a “no change in protein expression” was NOT confirmed with the data provided.
- In their previous publication in Scientific Reports the authors used β-tubulin as a loading control for GA treated cells. However, they did not show quantifications using β-tubulin. In general, would it be reasonable to discuss the use of β-tubulin as a loading control for WB in neurodegenerative diseases and in general?
- Did the authors study the effect of GA on polymerization of α-tubulin or α/β-tubulin?
Author Response

(The authors gave the same response as above.)

Round 2
Reviewer 1 Report
None
Author Response
Dear Dr. Yuan,
We are submitting a revised version of our manuscript (nutrients-912545) entitled " The effect of glyceraldehyde-derived advanced glycation end-products on β-tubulin-inhibited neurite outgrowth in SH-SY5Y human neuroblastoma cells." by Nasu et al.
We appreciate the helpful comments and suggestions made by the reviewers and editor. We have added the data which pointed out by reviewers to Fig. 2I and Fig. 3A, B, J in revised manuscript. We also carefully changed introduction, results including figures, legends, and discussion to help readers understand our results. All new text is shown navy.
We believe that our revised manuscript is now suitable for publication in Nutrients.
As the bands size is similar between β-tubulin and β-actin, we used the same lot samples with another loading membrane in this study. In Fig. 2I, we added the data of loading control. We also added the reasons to section 3.2. Line 9-11 in manuscript (Template Line No. 190-192).
The reviewer was worried about the non-specificity of antibodies of β-tubulin. The localization of β-tubulin immunoreactivity is in cytosol and/or axons. They appear to indicate the correct localization. At least, we can show a stained image of only the secondary antibody as negative control. As Fig. 3A and B, we added the data that non-specific and auto immunoreactivity could not be seen in SH-SY5Y cells using the only secondary antibody. These results supported that we could observed β-tubulin immunoreactivity in Fig.3C-E. We also added the explanation of this data to 3.3. section Line2-4 (Template Line No. 212-214). In the Fig. 3F-I and Fig.4D-G, we showed quantification for neurite outgrowth with immunocytochemistry by using α-internexin antibody because it is easy to observe neurites. We simply calculate neurite length of cells, not dependent on fluorescence intensity.
In 12 h treatments, the cell death of SH-SY5Y treated with RA and/or GA was evaluated using an MTT assay. RA and/or GA had not any effect on cell viability relative to the vehicle control. We also added the data of Fig. 3J and its results to 3.3. Line 14-16 (Template Line No. 224-226). And we added the discussion in Page 12 Line 23-24 (Template Line No.324-325).
Reviewer points out the specificity of antibody of TAGE in previous letter comment. According to reviewer’s comments, we added the explanation of TAGE-antibody specificity in discussion in Page 11 Line 20-24 (Template Line No. 276-280).
We added the sentences about the reasons and meanings, why we focused on β-tubulin protein for reader’s easier understanding in Page 2 Line28-30(Template Line No.70-72) in Introduction and Page 12 Line 19-25 (Template Line No.319-325).
Sincerely yours,
Yoshiki Koriyama, PhD
Associate professor
Graduate School and Faculty of Pharmaceutical Sciences
Suzuka University of Medical Science
3500-3 Minamitamagaki, Suzuka
Mie 513-8670, Japan

Round 3
Reviewer 1 Report
N/A